# The Impact of Diversified Farming Practices on Terrestrial Biodiversity Outcomes and Agricultural Yield Worldwide: A Systematic Review Protocol

**DOI:** 10.3390/mps4010008

**Published:** 2021-01-14

**Authors:** Andrea C. Sánchez, Natalia Estrada-Carmona, Stella D. Juventia, Sarah K. Jones

**Affiliations:** 1Bioversity International, Parc Scientifique d’Agropolis II, 34397 Montpellier, France; andrea.sanchez@cgiar.org (A.C.S.); n.e.carmona@cgiar.org (N.E.-C.); juventia.stella@wur.nl (S.D.J.); 2Farming Systems Ecology Group, Wageningen University & Research, 6700 AK Wageningen, The Netherlands

**Keywords:** agroecology, management practices, functional groups, biodiversity metrics, monoculture, natural habitats

## Abstract

The expansion and intensification of agriculture have led to global declines in biodiversity. This paper presents a systematic review protocol to clarify under what management and landscape contexts diversified farming practices are effective at improving outcomes for terrestrial biodiversity, and potential trade-offs or synergies with agricultural yields. The systematic review will be developed following the Reporting Standards for Systematic Evidence Syntheses (ROSES). The review will include articles that compare levels of diversity (e.g., abundance, richness, Shannon’s diversity index) of any terrestrial taxon (e.g., arthropods, mammals) in diversified farming systems to levels in simplified farming systems and/or natural habitats, prioritising articles that also report agricultural yields. We will search for relevant peer-reviewed primary studies in two global repositories: Scopus and Web of Science, and among primary studies included in previous meta-analyses that are retrieved from the search. Full-texts of identified articles will be screened using a clear inclusion/exclusion eligibility criteria. All included articles will be assessed to determine their internal validity. A narrative synthesis will be performed to summarize, describe and present the results, and where the articles provide sufficient and appropriate data, we will conduct a quantitative meta-analysis.

## 1. Introduction

Agriculture dominates approximately 40% of the world’s terrestrial surface [1]. The expansion of modern agriculture has caused the widespread conversion of diversified natural ecosystems into simplified monocultures with a small number and variety of cultivated plants [2]. These simplified systems are often highly dependent on external inputs and management practices, such as use of fertilizers and pesticides, irrigation, and tillage [3]. Since the last century, the expansion and intensification of agriculture has led to an increase of 23% in food production [4], while at the same time becoming one of the most important drivers of biodiversity loss worldwide [5,6]. With continued rapid population growth, global food demands are projected to double by 2050 [7], increasing the negative impacts on biodiversity [8]. Alternative farming practices are needed in order to promote livelihoods, food security and biodiversity conservation.

On-farm diversification practices have been proposed as suitable strategies to reduce the negative impact of modern agriculture on biodiversity while promoting sustainable food production [9,10,11,12]. Diversified farming systems involve the association of different varieties and/or species of plants, at multiple temporal and/or spatial scales, or integrating livestock and fish production with crop production [13,14]. Temporally diversified systems—such as systems under crop rotation—are recognised for increasing soil biodiversity responsible for nutrient recycling [15,16]. Plant diversification within and between agricultural plots—such as intercropping, agroforestry, hedgerows, and set-aside—can provide shelter, nesting sites and alternative food sources that promote the diversity of pollinators and natural enemies of pests [17,18,19]. Furthermore, at the landscape scale, the maintenance of natural and semi-natural habitats around and between agricultural fields positively influences the diversity of beneficial species [18,20,21,22]. Evidence is building that diversified farming can also have economic benefits by reducing input requirements and increasing or stabilising yields [14,23,24,25].

Previous reviews have synthesized the effects of diversified farming practices on biodiversity outcome metrics [21,26,27,28,29,30,31,32]. Others have summarised the effects on crop yields or yield stability [14,23,24,25,33,34,35,36]. Most of these reviews suggest that higher farm diversity has positive outcomes for biodiversity, yet that outcomes vary with several factors such as crop type, soil and agrochemical management, outcome metric (e.g., abundance, richness) and which taxa are assessed. The reviews also indicate that more diversified farms have more stable, and often higher, yields, but effects vary by crop, management, and agroecosystem context. While previous studies have greatly advanced our understanding, a new global meta-analysis of effects of diversified farming on biodiversity is valuable for several reasons. First, most of the syntheses focus on only a few taxonomic groups (e.g., arthropoda) [21,30,31] or broad taxonomic groups (e.g., vertebrate, invertebrate) [37]. More granular results are useful for conservation purposes, since diversity needs conserving at varietal, species and community levels. Second, few meta-analyses separate crop diversity from agrochemical or soil management effects, making it difficult to know whether domesticated plant diversity or management factors are responsible for positive outcomes on biodiversity [26,29,31,37,38]. Third, few meta-analyses compare diversified farming against natural habitats to help understand the potential contribution (and shortfall) of farm diversity in conserving biodiversity at levels found in natural habitat. Finally, there is a lack of clarity on when diversified farming can have positive effects on both biodiversity and crop yields to support both biodiversity conservation and food production goals. This is a gap that urgently needs closing to identify trade-offs and synergies and design appropriate incentives that enable farmers to rapidly transition towards more sustainable and biodiversity-friendly agriculture.

### Aim

The main objective of this systematic search is to provide clearer messages on when diversified farming practices are effective at improving outcomes for terrestrial biodiversity in specific agroecological contexts, while minimising trade-offs with agricultural yields. Publishing protocols for scientific reviews is a requirement of the Reporting Standards for Systematic Evidence Syntheses (ROSES) [39], yet rarely completed outside medical research. This systematic review protocol contributes to ensuring best standards for systematic reviews are followed in other fields, notably agronomy and ecology.

## 2. Protocol Design

The review protocol is a methodological framework for synthesizing evidence presented in primary studies. Such procedures aim to ensure the replicability, transparency and objectivity of searching, inclusion and synthesis of evidence in assessing specific research questions [40]. The systematic review will be developed following the ROSES guidelines [39] (see Figure 1). The completed ROSES’ checklist form is available in Appendix A.

## 3. Stakeholder Involvement

The review will be conducted as part of the CGIAR Water, Land and Ecosystems funded Sustainable foods through diversity-based practices (Sustainable Foods) project. Sustainable Foods will consult with biodiversity and food system experts including from Bioversity International (now part of the Alliance of Bioversity International and the International Centre for Tropical Agriculture), the United Nations Sustainable Development Solutions Network, Natural History Museum (UK) and Centre de coopération internationale en recherche agronomique pour le développement (France) to seek methodological guidance at key points in the review. For example, 10 stakeholders from across these institutions were convened in a workshop on 3–4 October 2019 to provide feedback and improve the research questions, scope, search strategy, database format, validation process and provisional analysis strategy. These experts will be invited to review and contribute to the peer-reviewed papers presenting the final methods and results.

## 4. Step 1: Research Questions

### 4.1. Primary Question

How effective is diversified farming at improving outcomes for terrestrial biodiversity and what are the trade-offs or synergies with yield?

#### Components of the Primary Question

The PICOC (Population, Intervention, Comparator, Outcomes, Context) framework [41] for the primary study question are detailed as follows:Population: terrestrial biodiversity in diversified and simplified farming systems and/or natural habitats worldwide, such as bacteria, animals, fungi, plants, etc.;Intervention: diversified farming practices (e.g., agroforestry, crop rotation, intercropping);Comparator: simplified farming practices (e.g., monoculture), and/or natural habitats (e.g., natural grasslands, primary or secondary forest);Outcomes: biodiversity outcome metrics (e.g., abundance, species richness, Shannon’s diversity index, percentage fungi colonization, etc.) and crop yield (i.e., kg/ha);Context: primary field-based studies with experimental and observational designs conducted anywhere in the world in agricultural fields, with or without comparable studies conducted in natural habitats.

### 4.2. Secondary Questions

What is the state of evidence of diversified farming practices on biodiversity outcomes and trade-offs or synergies with yield, in terms of number of studies, geographic coverage, intervention types, population types (taxa group, functional group), outcome metrics, crop types, intervention practices?Do the effects of diversified farming on biodiversity outcomes and crop yield vary across different:
taxonomic groups (e.g., bacteria, fungi, mammals),functional groups (e.g., pest, pollinators, decomposers),outcome metrics (e.g., abundance, species richness, Shannon’s diversity index),crop type (e.g., nuts, vegetables, fruits, etc.),management practices (i.e., fertilizer application, pesticide application, soil management),diversified farming practice (e.g., agroforestry, crop rotation, intercropping).


## 5. Step 2: Searching for Relevant Articles

The search strategy aims to identify a wide number of relevant articles published in peer-reviewed journals. We will not include studies reported in grey literature. The article’s search will be accomplished in three different stages. First, we will search in Scopus (https://www.scopus.com/) and Web of Science (https://apps.webofknowledge.com/), using search strings to identify potentially relevant articles, restricting the search to titles, abstract and keywords in English language articles (Table 1). The search string has been developed in consultation with scientists working on diversified farming practices and/or biodiversity outcomes, convened in October 2019 through the CGIAR-funded Sustainable Foods project. The string has been constrained to search for articles that explicitly consider gradients of farm or landscape diversity, and trade-offs or synergies with yield, to limit the number of non-relevant articles retrieved (see Appendix B). Secondly, we will expand the search of articles by extracting the reference list from all meta-analyses found during the preliminary searches. Third, we will include relevant primary studies known to scientists consulted through the Sustainable Foods project and which are not picked up by the search string. The last two supplementary search processes will help us to identify and include all relevant primary studies including those included in previous related meta-analyses to build on existing knowledge. 

We will assess the comprehensiveness of our search by comparing the number of articles retrieved against the articles retrieved and included in previous, similar reviews. We will compile the CSV export of all the document results from all search stages, and remove any duplicates based on the DOI and article title. All remaining articles will be assigned a unique ID (Study_ID). The search will be updated if it was performed more than 12 months prior to submission of the final results to a peer-reviewed journal.

## 6. Step 3: Selection of Articles

### 6.1. Screening Process

Portable document format (PDF) versions of all accessible articles retrieved during the search stage will be downloaded and renamed with the corresponding Study_ID. Authors are affiliated with Bioversity International, Imperial College London, Wageningen University and Research, and King’s College London. We will download only articles that are open access or accessible through subscriptions made by these institutions. The articles will be screened for inclusion at full-text level against our eligibility criteria by two independent reviewers. The articles will be classified as: (1) include, (2) exclude, and (3) maybe. The decision to include the articles classified as “maybe” will be taken in consensus with a third reviewer based on the eligibility criteria. All excluded articles will be coded with an exclusion reason.

### 6.2. Eligibility Criteria

Only articles published in peer-reviewed journals with full-text in English will be included in this review. Where articles report secondary data (i.e., data from another study), we will exclude the article. The following inclusion criteria will be applied, described using the five PICOC components:Eligible Populations: any non-domesticated terrestrial micro or macro-organisms. This includes any surveyed organisms or group of organisms that can be classified to taxonomic phylum level, order or below (i.e., family, species level). Any life stage will be considered.Eligible Interventions: the articles must provide a clear description of the diversified agricultural systems. Diversified farming systems are represented by the association of different plant species (e.g., two crops; a crop and a beneficial non-crop), or different varieties/cultivars/accessions of crops (e.g., two crop genotypes) at multiple temporal and/or spatial scales, or the integration of livestock and fish production with crop production. We will distinguish six types of diversified farming systems as described in Table 2.Eligible Comparators: articles must provide enough information to differentiate the intervention systems to the comparators. We will classify the simplified farming systems assessed into monoculture and simplified others, and natural habitats into two types of habitats natural habitats, and abandoned farmland (see Table 2). We will not include outcomes from tree plantations for timber or any other commercial purposes.Eligible Outcomes: articles must report a quantitative assessment of the effect of interventions and comparators on outcomes for non-domesticated terrestrial biodiversity. Outcomes may include metrics such as: abundance, species richness, activity-density, Chao1 index, colonization percent, Fisher alpha, Jaccard similarity index, Jack-knife species richness, Margalef Index, number of orders, Pielou index, rarefied species richness, Shannon–Wiener index, Shannon evenness index, Shannon index, Simpson index, Simpson’s reciprocal index, Simpson index, species evenness, reproductive success. Included articles must report location data (e.g., geographic coordinates, country), outcome means (or medians), sample sizes, and variance measures (e.g., standard deviation, standard error, interquartile ranges, confidence intervals) for interventions and comparators assessed. The biodiversity outcomes at intervention and control sites must be comparable, i.e., data collected at the same or very similar points in time, using the same or very similar sampling methods.Context: we will include only data from primary field-based studies with experimental and observational designs conducted in natural habitats, agricultural fields on farms or outdoor experimental research sites. We will exclude primary studies carried out in laboratories or greenhouses.

The eligibility criteria may be adapted during the articles screening process to overcome any shortcomings that emerge, e.g., to include different biodiversity outcomes, variance measures; and/or other intervention or control systems.

## 7. Step 4: Data Extraction and Coding Strategy

Qualitative and quantitative data will be extracted from all articles that fulfil all the inclusion criteria. The extracted data will be recorded in a contiguous database specially designed and coded for this review (see Appendix C). Intervention and comparator observations from the same study will be recorded and coded in different individual rows, linked by study (Study_ID) and experiment (Control-Intervention_ID) unique identifiers. This database structure will help to correctly record the data when articles assess multiple intervention or comparators, or more than one taxonomic group or biodiversity metric. Including multiple observations when they are provided in a single article will allow us to use all available data and thus estimate more precise effect sizes than using only a single pair of observations from each article [40]. We provide clear explanations of how we are going to record and code the extracted data in Appendix C. One reviewer will enter the data and a second reviewer will validate the accuracy and completeness of the data entry, for at least 80% of entries. If there is any disagreement between reviewers, the review team will discuss these to find agreement and make modifications to the database as necessary.

The extracted data will be used to assess the primary and secondary questions established. We will extract qualitative data on bibliography information (e.g., authors, publication year, title); Study_ID (i.e., numeric ID associated with each unique article, assigned during the screening process); Control-Intervention ID (i.e., numeric ID associated with each unique intervention or comparator, assigned during the coding process); dominant crop common name (e.g., maize, rice, banana); Experiment ID (i.e., numeric ID associated with each intervention-comparator comparison group, assigned during the extraction process to identify e.g., matching sampling dates in a repeat sampling design, matching study location); crop type (common name, scientific name, and Food and Agricultural Organisation of the United Nations commodity group); agricultural system as intervention or comparator (e.g., intercropping, monoculture, agroforestry, crop rotation, set aside); common and scientific name of the taxa sampled (e.g., ants, Formicidae); functional group sampled if specified (e.g., pest, decomposers; predator); biodiversity outcome metric (e.g., species richness, abundance, Shannon’s diversity index); the sampling method (e.g., transect, trap), pesticide use (yes or no, and kg/ha), fertilizer use (yes or no), fertilizer chemical (yes or no), soil management (e.g., tillage, no tillage, slash and burn); landscape characteristics (e.g., % agricultural land use, climate); and location of the study (i.e., country/state and geographical coordinates).

The quantitative outcome data that will be recorded include: biodiversity outcome means; biodiversity outcome variance measure (e.g., standard deviation, standard error of the mean, interquartile range, confidence intervals); number of samples collected; crop yield mean (e.g., kg/ha, g/plant), crop yield variance (e.g., standard deviation, standard error of the mean); farm size (in hectares); length of time that the land has been in current state (in years); and duration of the sampling period (in days, from start to finish). Data on biodiversity outcomes and crop yield will be extracted directly from publication text, tables, figures, or supplementary information. Data from figures will be extracted using GetData Graph Digitizer 2.26 or WebPlotDigitizer v4.2. Where data values or units are unclear or not provided (e.g., the meaning of the error bars in figures), the authors of the corresponding article will be contacted by email. If the authors do not respond, the data entry will be removed.

We will extract the data from all interventions and comparators if an article reports outcomes for multiple interventions (e.g., agroforestry and intercropping), or multiple comparator systems (e.g., simplified agricultural systems and natural). When outcomes are presented for multiple years (and no average across years is provided), we will include only the data of the last year assessed; otherwise, we will include the average across years. When outcomes are presented for multiple crop stages (e.g., flowering and non-flowering) or survey days (e.g., 25 and 55 days after sown) for the same year, and the average from across the study is not provided, we will extract all of the outcomes separately. When outcomes are reported at different distances away from the sampling plot, we will only include the data collected closest to the sampling plot. When biodiversity outcomes are reported for multiple taxa groups (e.g., ants, birds, and bees), functional groups (e.g., pests and decomposers), sample methods (e.g., vacuums and pitfall traps), and/or locations, we will enter each one separately in the database. When studies report outcomes for different life stages (e.g., adult, larvae, pupae), we will record only the most advanced stage. In the case where outcomes are disaggregated by functional group, and further disaggregated by taxonomic group, we will record the outcomes by functional group only. When a study presents multiple geographical coordinates for the same intervention, comparator, taxa, functional group or biodiversity measure, and these points are in the same region, we will record geographic coordinates of the centroid only. If a study does not present the exact location of the study area (i.e., in geographical coordinates), we will use the description of the study location to identify the proximate geographic coordinates, making sure to locate the points in agricultural or natural areas as described in each article.

After the data extraction process, we will reorganize the database using an R script so that each row of data contains observations from one comparator and an associated intervention from the same article. This procedure will facilitate the calculation of effect sizes.

## 8. Step 5: Observations Validity Assessment

Each observation (i.e., comparison between outcomes from one comparator and an associated intervention) of all included articles will be assessed to determine its internal validity (i.e., the probability to present bias). We will not consider external validity (i.e., how generalizable the observation is) as this is likely to vary geographically and with the population and intervention/comparator assessed.

Table 3 includes the list of criteria for the observation validity assessment. These criteria represent what we will consider relevant to rate the quality of the included observations, based on objective measures of internal validity. Sources of potential bias accounted or corrected for through meta-analytical procedures are excluded from Table 3 (e.g., different sample sizes between interventions and comparators; non-independence of effect sizes within studies).

Each observation will be scored as having a “Low”, “High”, or “Unclear” risk of bias relating to each of the criterion in Table 3. Observations that present “High risk” and/or “Unclear” for two or more assessment criteria will be classified as having a high risk of bias. All the necessary data to score each observation against the validity criteria will be recorded during the data extraction process. The quality and consistency of the recorded information will be checked as described in the data extraction and coding section.

## 9. Step 6: Identification of Potential Effect Modifiers/Reasons for Heterogeneity

The review team will extract the potential effect modifiers directly from the included studies or will re-classify the extracted data to generate the information. The following list details the factors (moderators) that will be considered as potentially causing effect size variation between and within the included studies:Crop typePopulation taxonomic groupFunctional groupType of intervention practice (i.e., type of diversified system)Biodiversity outcome metricFertilizer applicationPesticide applicationSoil managementLandscape context (i.e., composition, structure)

Additional effect modifiers may be included to this list as the review proceeds. The list of potential effect modifiers was compiled based on evidence reported in the literature including in previous related meta-analysis [26,37,38].

## 10. Step 7: Data Synthesis and Presentation

Results of the data analysis will synthesize evidence of farm diversity, pesticide and fertiliser management effect on a variety of taxa, functional groups, and across biodiversity metrics. Additionally, the analysis will also quantify the effect of different farm diversification strategies on agricultural yields. A narrative synthesis will be performed in order to summarize, describe and present the results. A descriptive statistic using figures and tables will be conducted to visualize the distribution of the data across countries, taxa groups, functional groups, intervention practices, crop types and quality of observations (assessed against our internal validity criteria).

We will use effect sizes based on means to conduct a quantitative meta-analysis of the impact of interventions and comparators on biodiversity outcomes and yield. We will calculate the effect sizes using: (i) simplified farming systems as the control and diversified farming systems as the intervention, or (ii) natural habitat as the control and diversified farming systems as the intervention. Then, we will conduct the meta-analyses in R-4.0.0 [42] using the package *metafor* [43]. We will perform random-effect meta-analysis, to account for between-study and within-study variance [44]. Effect sizes will be pooled using multi-level mixed effects models to obtain overall mean effect sizes. Meta-regression procedures will be performed to examine the moderator effect of potential modifiers on biodiversity outcomes. The mean effect of intervention/control systems and the variance values from the meta-analysis will be shown using forest plots.

The presence of publication bias will be identified and assessed using visualization and statistical methods proposed by Nakagawa and Santos [45] for biological meta-analyses. We will use Cook’s distance or hat values and other established methods to identify potential influential cases and extreme observations [46]. We will conduct sensitivity analysis to compare the results from the meta-analytical models fitted with all the included studies and the models fitted after excluding outliers. All data and codes used in the analysis will be made available on publication of the results.

The compiled database of effect sizes will be made publicly available in raw format and through an interactive and user-friendly website which provides a comprehensive overview of the effects of diversified farming systems on terrestrial biodiversity outcomes and yield. We will also provide the literature included in the analysis for transparency. This would assist researchers and policymakers to assess the effects of different strategies and design improved interventions for their country, crop, farming system, or taxa of interest.

## Figures and Tables

**Figure 1 mps-04-00008-f001:**
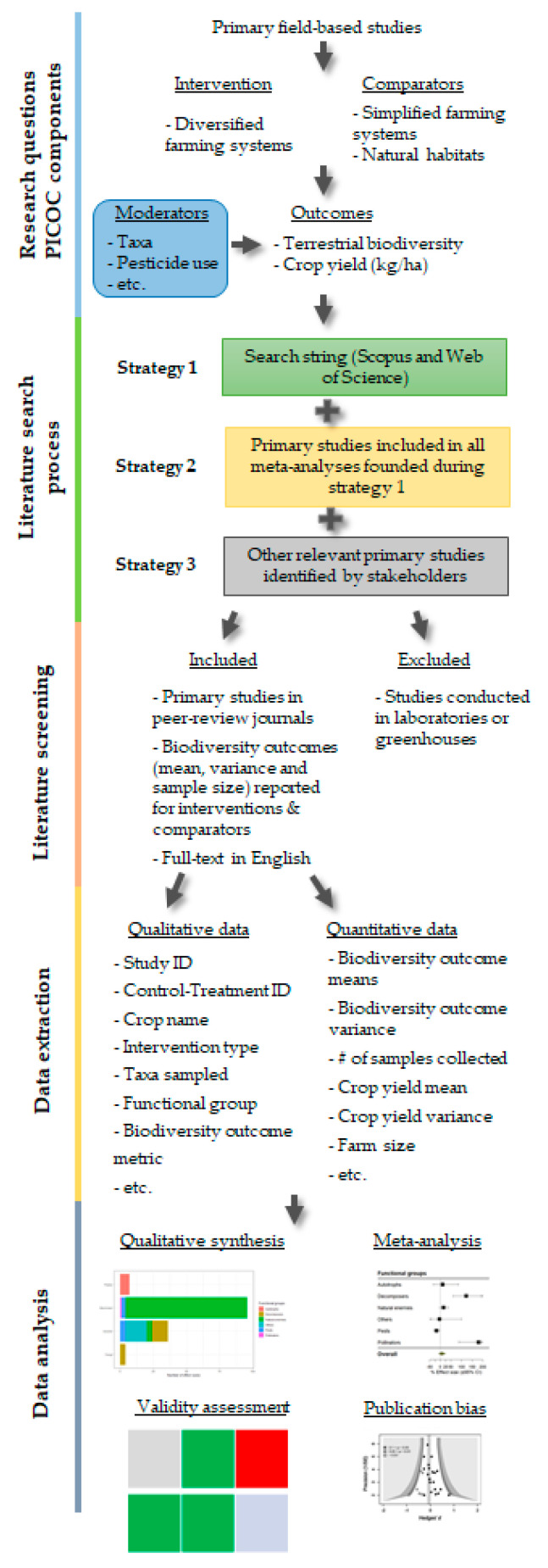
Systematic review protocol methodological steps following Reporting Standards for Systematic Evidence Syntheses (ROSES) guidelines.

**Table 1 mps-04-00008-t001:** Search strings that will be used to identify relevant articles in two global repositories. The wildcard “*” is used to represent unknown characters or no character, and “?” to represent a single character or no character, to allow for variable spellings and truncations of search terms.

Repository	Search String
Scopus	TITLE-ABS-KEY (“agricultur*” AND “biodiversity”) AND TITLE-ABS-KEY (“agro?ecology” OR “agro?biodivers*” OR “agroforestry” OR “border plant*” OR “riparian buffer” OR “woodlot” OR “hedgerow” OR “cover crop*” OR “crop rotation” OR “crop divers*” OR “inter?crop*” OR “mixed crop*” OR “cultivar mixture” OR “plant divers*” OR “polyculture” OR “tree divers*” OR “variet* diversity” OR “fallow” OR “field margin*” OR “grass strip*” OR “*flower strip*” OR “insect* strip” OR “conservation strip” OR “vegetation strip” OR “catch crop” OR “inter?crop*” OR “crop variety” OR “crop sequenc*” OR “mixed farming” OR “land sparing” OR “landscape heterogeneity” OR “heterogeneous landscape” OR “landscape diversi*” OR “divers* landscape” OR “homogeneous landscape” OR “landscape homogeneity” OR “landscape complexity” OR “simplif* landscape” OR “complex landscape” OR “multi?function* landscape” OR “integrated crop-livestock” OR “integrated crop-forest” OR “land sharing”) AND TITLE-ABS-KEY (“ richness” OR “ abundance” OR “species diversity” OR “functional diversity” OR “index”) AND TITLE-ABS-KEY (“crop yield” OR “crop production”) AND (LIMIT-TO (LANGUAGE, “English”)).
Web of Science	Web of Science search string: (TS= (“agricultur*” AND “biodiversity”) AND TS= (“agro?ecology” OR “agro?biodivers*” OR “agroforestry” OR “border plant*” OR “riparian buffer” OR “woodlot” OR “hedgerow” OR “cover crop*” OR “crop rotation” OR “crop divers*” OR “inter?crop*” OR “mixed crop*” OR “cultivar mixture” OR “plant divers*” OR “polyculture” OR “tree divers*” OR “variet* diversity” OR “fallow” OR “field margin*” OR “grass strip*” OR “*flower strip*” OR “insect* strip” OR “conservation strip” OR “vegetation strip” OR “catch crop” OR “inter?crop*”OR “crop variety” OR “crop sequenc*” OR “mixed farming” OR “land sparing” OR “landscape heterogeneity” OR “heterogeneous landscape” OR “landscape diversi*” OR “divers* landscape” OR “homogeneous landscape” OR “landscape homogeneity” OR “landscape complexity” OR “simplif* landscape” OR “complex landscape” OR “multi?function* landscape” OR “integrated crop-livestock” OR “integrated crop-forest” OR “land sharing”) AND TS= (“richness” OR “abundance” OR “species diversity” OR “functional diversity” OR “index”) AND TS= (“crop yield” OR “crop production”)) AND LANGUAGE: (English)

**Table 2 mps-04-00008-t002:** Control and intervention systems descriptions.

System	Description	Type
Agroforestry	Following Beillouin et al. [27], agroforestry satisfies three conditions: (i) at least two plant species interact biologically, (ii) at least one of the plant species is a woody perennial, and (iii) at least one of the plant species is managed for forage, annual or perennial crop production. Includes alley cropping with trees, shade monoculture, silvo-pasture.	Intervention
Cover crops	Following Beillouin et al. [27], plant grown in addition to the main crop for agronomic purposes, e.g., to manage soil erosion, pests, soil fertility or soil quality. The associated plant could be harvested or not, perennial or not.	Intervention
Crop rotation	Following Beillouin et al. [27], recurrent succession of a set of selected crops grown on a particular agricultural land each season or each year according to a definite plan.	Intervention
Diversified other	Diversity-based practices not included in other categories. Includes combinations of single practices, such as crop rotation and cover crops used in unison, and integrated crop-livestock systems.	Intervention
Embedded natural	Land on-farm not used for farming and where non-crop plants are sown or regenerated naturally to benefit biodiversity or for other environmental purposes. Includes fallow (regular, >6 months), field margins, hedgerows, riparian buffers, set aside, vegetation strips, flower strips.	Intervention
Intercropping	Adapted from Beillouin et al. [27], the simultaneous cultivation in the same field of two or more crop species, varieties, or cultivars, for all or part of their growth cycle. All crops are harvested.	Intervention
Monoculture	The cultivation of a single crop species or variety in the same plot at the same time or continually in different seasons.	Control
Simplified other	Relatively low diversity (usually only 2 species) agroforestry, cover crop, crop rotation or intercropping, for studies comparing these against the same cropping system planted with relatively high diversity (usually 3 or more species). Also used for cropped areas with no embedded natural features (e.g., hedgerows, vegetation strips) when compared against cropped areas with these embedded natural natures.	Control
Abandoned farmland	Abandoned cropland left to rewild.	Control
Natural	Forests, shrubland or grassland that is not commercially harvested and which, if managed, is managed for conservation purposes. Can include primary or secondary vegetation growth. Includes fynbos, natural or semi-natural grassland, remnant vegetation, secondary successional habitat.	Control

**Table 3 mps-04-00008-t003:** List of criteria for observation validity assessment.

Assessment Criteria	Low Risk of Bias	High Risk of Bias	Unclear Risk of Bias	Type of Bias Addressed
1. Intervention and comparator sample size	Total sample size >= 5	Total sample size < 5	-	Selection bias
2. Are the interventions and comparators matching at the same site (i.e., same climate conditions, weather events, soil type)	Yes	No	Insufficient data to permit assessment of ‘Low risk’ or ‘High risk’ bias	Selection bias
3. Time the interventions and comparators plots have been in this state	>=1 year	<1 year	Insufficient data to permit assessment of ‘Low risk’ or ‘High risk’ bias	Selection bias
4. Can the intervention clearly be classified?	Yes. The intervention can be classified as one of the diversified systems specified in Table 2	No. The intervention was described by the article as “polyculture”.	-	Selection reporting, performance bias,

## Data Availability

No new data were created or analyzed in this study. Data sharing is not applicable to this article.

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
