# Peer review of "The Impact of Diversified Farming Practices on Terrestrial Biodiversity Outcomes and Agricultural Yield Worldwide: A Systematic Review Protocol"

_mps, 2021, doi:10.3390/mps4010008_

Round 1
Reviewer 1 Report
The work does not present the structure of a scientific article. The objectives are confusing. The methodology and results are also unclear. There is no discussion or conclusions. The text must be presented in a clearer way in order to be evaluated.
Author Response
Response to Reviewer 1 Comments
Point 1: The work does not present the structure of a scientific article. The objectives are confusing. The methodology and results are also unclear. There is no discussion or conclusions. The text must be presented in a clearer way in order to be evaluated.
Response 1: Publishing protocols for systematic review is still quite unusual in agronomy or ecology, but a requirement for RepOrting Standards for Systematic Evidence Syntheses (ROSES) guidelines, and common practice in medical research. Therefore, we understand the reviewer is not used to seeing articles in this format, but would urge him/her to support the publication of protocols for systematic reviews outside of medical research, to help other fields achieve better standards in conducting reviews. We have followed the protocol structure and content proposed by ROSES. Also please see the appendices which provide the ROSES check list form (justifying our article layout), and full detail on the search methodology, data coding and study. We have now clarified in the Aim the study the purpose of our systematic review protocol: lines 80-83 - “Publishing protocols for scientific reviews is a requirement of the RepOrting Standards for Systematic Evidence Syntheses (ROSES)[39], yet rarely completed outside of medical research. This systematic review protocol contributes to ensuring best standards for systematic reviews are followed in other fields, notably agronomy and ecology.”
Reviewer 2 Report
I agree with the authors that there is clear value in this review. I also appreciate the thoroughness of their methods. I have a few specific suggestions below.
Line 102 – Only in agricultural fields? But what about natural or restored habitats adjacent to the fields? Or agroforestry systems like windbreaks? Or the embed natural in Table 2.
Line 107 – I recognize this is not an exhaustive list, but what about evenness. Or reproductive success. Similarly in lines 119-124. Perhaps these could be a table? I see now that this is at line 179. But reproductive success is still lacking. This is key to avoid the suggestion of an ecological trap.
Table 1 – Could I suggest adding specific taxonomic groups to the keywords for biodiversity. For example; plant, bird, mammal, or insect. Many papers are titled to be more specific than just generic biodiversity. Likewise, pairing species with abundance within the search is going to limit taxa or species-specific titles/abstracts.
Line 153-154 – this seems like an odd limiting factor. How much effort will be made to make sure this does not create a bias? Same with line 160.
Table 2 – what about organic? Or regenerative?
Line 290 – I appreciate the acknowledgment that others will be added. I would add landscape context as a very important modifier. For example here - Burel, F., Butet, A., Delettre, Y.R. and De La Peña, N.M., 2004. Differential response of selected taxa to landscape context and agricultural intensification. Landscape and Urban Planning, 67(1-4), pp.195-204.
Author Response
Response to Reviewer 2 Comments
Reviewer 2:
Point 1: I agree with the authors that there is clear value in this review. I also appreciate the thoroughness of their methods. I have a few specific suggestions below. Line 102 – Only in agricultural fields? But what about natural or restored habitats adjacent to the fields? Or agroforestry systems like windbreaks? Or the embed natural in Table 2.
Response 1: Text in Lines 110-111 “Population: Terrestrial biodiversity in agricultural fields worldwide, such as bacteria, animals, fungi, plants, etc.” replaced by “Population: Terrestrial biodiversity in diversified and simplified farming systems and/or natural habitats worldwide, such as bacteria, animals, fungi, plants, etc.”
Point 2: Line 107 – I recognize this is not an exhaustive list, but what about evenness. Or reproductive success. Similarly, in lines 119-124. Perhaps these could be a table? I see now that this is at line 179. But reproductive success is still lacking. This is key to avoid the suggestion of an ecological trap.
Response 2: Line 193 – “reproductive success” added to the list of biodiversity outcomes.
Point 3: Table 1 – Could I suggest adding specific taxonomic groups to the keywords for biodiversity. For example; plant, bird, mammal, or insect. Many papers are titled to be more specific than just generic biodiversity. Likewise, pairing species with abundance within the earch is going to limit taxa or species-specific titles/abstracts.
Response 3: Thank you for this suggestion. We agree with the suggestion to remove ‘species’ and have amended Table 1 to replace “species abundance” and “species richness” by “abundance” and “richness” in the Scopus and Web of Science search strings. While we acknowledge that naming individual taxon in the search strings may identify some additional studies, the list of taxon we could include is almost limitless. We have found that most articles draw out wider implications to biodiversity in the abstract and so we consider the number of omitted articles will be minimal and prefer to keep the number of articles that need to be screened to manageable numbers. A preliminary search shows that our search process retrieves >1100 articles, which is comparable to previous similar reviews.
Point 4: Line 153-154 – this seems like an odd limiting factor. How much effort will be made to make sure this does not create a bias? Same with line 160.
Response 4: Articles that are not accessible using existing institutional subscriptions will not be included as we do not have project funding to pay for individual article access, while the idea to include only articles published in peer-review journals is to avoid possible low-quality studies that did not pass for a peer-review process. In the lines 324 - 327 we declared our commitment to carry out publication bias assessment to the meta-data.
Point 5: Table 2 – what about organic? Or regenerative?
Response 5: We do not include these terms since organic agriculture is defined in different ways in different countries, and usually includes avoiding use of chemical fertilizers and pesticides, but not necessarily increasing crop or farm-level diversity. Likewise, regenerative agriculture does not necessarily involve increasing crop or farm-level diversity.
Point 6: Line 290 – I appreciate the acknowledgment that others will be added. I would add landscape context as a very important modifier. For example, here - Burel, F., Butet, A., Delettre, Y.R. and De La Peña, N.M., 2004. Differential response of selected taxa to landscape context and agricultural intensification. Landscape and Urban Planning, 67(1-4), pp.195-204.
Response 6: Introduction lines 50-52: We added the sentence “Furthermore, at the landscape scale, the maintenance of natural and semi-natural habitats around and between agricultural fields positively influences the diversity of beneficial species (Karp et al., 2013; Garibaldi et al., 2014; Gontijo, 2019; Burel et al., 2004).”.; Line 302 – Landscape context (i.e., composition and structure) added to the list of possible modifiers.
Reviewer 3 Report
The paper proposes a protocol for a systematic review on the impact of diversified farming practices on terrestrial biodiversity outcomes and agricultural yield worldwide.
Obviously, this is only a proposal and there are no data at the moment that can support the effectiveness of the proposed protocol, so problems that may arise in reaching the goal, particularly with regard to the possibility of systematic biases between reference and treatment stands should be specified.
Lines 64-70 : please, report here the related literature.
Line 222: amount and type of fertilizer?
Author Response
Response to Reviewer 3 Comments:
Point 1: The paper proposes a protocol for a systematic review on the impact of diversified farming practices on terrestrial biodiversity outcomes and agricultural yield worldwide. Obviously, this is only a proposal and there are no data at the moment that can support the effectiveness of the proposed protocol, so problems that may arise in reaching the goal, particularly with regard to the possibility of systematic biases between reference and treatment stands should be specified.
Response 1: We will only include data from literature that clearly explain the intervention and comparator systems (please see lines 180-181 in the Eligibility criteria section), so we do not consider it likely that there will be systematic biases other than those identified in the validity assessment section. However, if during the implementation of this protocol we uncover other possible sources of bias we will adapt our methodology. We have added a sentence in line 198-200 to explain this as follows: “The eligibility criteria may be adapted during the articles screening process to overcome any shortcomings that emerge, e.g., to include different biodiversity outcomes, variance measures; and/or other treatment or control systems.”
Point 2: Lines 64-70: please, report here the related literature.
Response 2: Line 68 – Related literature added (Tuck et al., 2014; Gonthier et al., 2014; Venter et al., 2016; Letourneau et al., 2011; Shackelford et al., 2013)
Point 3: Line 222: amount and type of fertilizer?
Response 3: Line 234: Added: fertilizer chemical (yes or no).